# Comparing Reproductive Health Awareness, Nutrition, and Hygiene among Early and Late Adolescents from Marginalized Populations of India: A Community-Based Cross-Sectional Survey

**DOI:** 10.3390/healthcare9080980

**Published:** 2021-08-02

**Authors:** Shantanu Sharma, Faiyaz Akhtar, Rajesh Kumar Singh, Sunil Mehra

**Affiliations:** 1Department of Clinical Sciences, Lund University, Skåne University Hospital, S-20502 Malmö, Sweden; 2MAMTA Health Institute for Mother and Child, Delhi 110048, India; akhtar@mamtahimc.in (F.A.); rksingh@mamtahimc.in (R.K.S.); dr_mehra@mamtahimc.in (S.M.)

**Keywords:** anemia, growth and development, menstruation, rural population, sanitation

## Abstract

Adolescence (10–19 years) is marked by many physiological changes and is vulnerable to health and nutritional problems. Adolescence, particularly, early adolescence is inadequately addressed in our national surveys. The present study aimed to assess the reproductive health awareness, nutrition, and hygiene of marginalized adolescent girls and boys and compare them among early and late adolescents. Our cross-sectional study was a part of a community-based project across India’s five zones, namely North, East, West, Central, and South. Unadjusted and adjusted logistic regression was performed to compare awareness about HIV/AIDS, or Sexually Transmitted Infections (STI), consumption of Iron Folic Acid (IFA) tablets and three meals in a day, safe menstrual hygiene practices, history of anemia, and open defecation practice among early and late adolescents. Data were reported as unadjusted and adjusted odds ratio (aOR) with a 95% confidence interval (95% CI). Among early adolescents, around 58% of girls and boys did not consume IFA tablets, and 28% of girls and 24% of boys defecated in the open. Compared to late adolescents, early adolescent girls had lower odds of awareness about HIV/AIDS (aOR (95% CI): 0.50 (0.47–0.54)) and open defecation (aOR (95% CI): 0.90 (0.83–0.98)) and higher odds of hand hygiene after defecation (aOR (95% CI): 1.52 (1.37–1.68)) and safe menstrual practices (aOR (95% CI): 1.42 (1.23–1.64)). There is a dire need to start public health interventions from early adolescence for long-term benefits throughout adolescence.

## 1. Introduction

Adolescence (10–19 years) is a critical phase in the life course, as a phase of enormous physical, social, and cognitive development. Adolescence is frequently dichotomized into two stages, early (10–14 years) and late adolescence (15–19 years) [1]. Early adolescence begins with a growth spurt and the development of sex organs and secondary sexual characteristics. The major physical development has already taken place by late adolescence. Although enormous physiological development in the brain and risk-taking behavior starts in early adolescence, it continues in late adolescence. Late adolescence is seen as a phase of opportunity, idealism, and promise [2].

India is home to 243 million adolescents, constituting approximately one-fifth of the total population of the country. This large population of adolescents represents a demographic dividend that has the potential to benefit India socially, politically, and economically. To reap these benefits, adolescents should be healthy, educated, and skilled [3]. However, several studies highlighted vulnerabilities and challenges adolescents in India are facing. Around 29% of boys and 58% of girls between 15–19 years are anemic in India. Similarly, 42% of girls and 45% of boys are thin (Body Mass Index < 18.5 kg/m^2^). Less than half of girls or boys consume green vegetables daily, and 50–52% of girls and boys consume fruits occasionally or never. Besides, a large proportion of girls are married before 18 years of age in India [4,5].

Open defecation and lack of hand hygiene are common among adolescents in India [6,7,8]. Previous studies demonstrated that not only is there a high burden of anemia but also low compliance of Iron Folic Acid (IFA) consumption among adolescents in India. To address this public health problem of anemia, the government of India launched the *Anemia Mukt Bharat* program that includes weekly iron-folic acid supplementation of adolescents in schools or *Anganwadi* Centers for out-of-school adolescents [9]. Regular food consumption is poor among adolescents, with 45% missing their regular meals once or more often in a week [10].

Awareness about HIV/AIDS or Sexually Transmitted Infections (STI) is limited among adolescents, as revealed in the previous studies [11,12]. Besides lack of awareness, girls do not practice safe menstrual hygiene practices. The use of cloths is widely prevalent among Indian women (around 62%), and their sanitization with soap and water is poorly performed [4]. Furthermore, menstruation in India is surrounded by many taboos and social stigma that restrict women and girls from talking about it openly and resort to safe menstrual hygiene practices [13].

Among all social classes, marginalized populations (scheduled castes, or tribes, or other backward classes) are more vulnerable to poor health practices, nutritional intakes, and hygiene maintenance [14,15]. This is evident from the national surveys reporting 7% teenage pregnancies among tribal populations compared to 1% among non-tribals [16]. Tribals constitute approximately 9% and scheduled castes about 16.6% of the total population of India and are socio-economically disadvantaged and disproportionately affected by limited access to health services [17,18]. Studies found poor health and nutrition indicators, such as lower dietary diversity, higher prevalence of anemia, and limited awareness about HIV/AIDS among adolescents belonging to scheduled castes and tribes compared to non-marginalized groups [19,20].

Most of the studies explored the health, nutrition, and hygiene status of late adolescents, and there has been limited literature for early adolescents across low and middle-income countries. Smaller sample sizes, limited geographic coverage and community-based studies, and missing early versus late adolescents’ comparison are some of the lacunae in the previous studies. There is emerging evidence for investing in early adolescence research and interventions to encourage the development of positive health that is transformative over the course of life [21]. Considering the need to address the lack of data and baseline data to support intervention strategies for improving key health, nutrition, and hygiene indicators of adolescents in India, we proposed to do the study. The present study aimed at assessing the reproductive health awareness, nutrition, and hygiene situation of adolescent girls and boys and comparing them between early and late adolescents.

## 2. Materials and Methods

### 2.1. Participants and Procedures

The study was a part of a community-based project, and the data were collected to assess the health, nutrition, and hygiene situation of adolescents among marginalized populations in India. The participants were recruited from each of the five zones of India, namely North (Rajasthan, Chandigarh, and Delhi), East (Bihar), West (Maharashtra), Central (Uttar Pradesh), and South (Karnataka). One or two district(s) per state in each of the five zones were selected, and two blocks per district were chosen randomly. In each block, villages/wards were ranked based on the percentage of marginalized populations (scheduled caste, scheduled tribes, other backward classes, and economically weaker sections). Around 120–140 villages per block with the highest proportion of marginalized populations were selected. The frontline workers in every village/ward were contacted, and their list of households with adolescents (girls and boys) was referred. In a household, one girl and/or boy was selected for the interview. If the girl or boy was not found, a second visit was not made and a subsequent household was selected. We interviewed unmarried girls and boys (10–19 years) living in the study area for at least one year.

This cross-sectional study was conducted in 2017 and 2018. The data were collected by a team of 5–6 investigators per district. All the data investigators were female and trained through a two-day training session, including simulation exercises. The ethical clearance was granted by the MAMTA Ethical Clearance committee (MERB/Mar-2017/002). All the interviews were conducted in the local language (Hindi). Consent was obtained from the parents of adolescents and assent from adolescents for participation in the study.

### 2.2. Measures

The quantitative questionnaire consisted of questions related to socio-demographic data, including age, gender (male/female), religion (Hindu or non-Hindu), and caste (marginalized, including scheduled castes or tribes or other backward classes; or non-marginalized). Furthermore, questions about the education status (illiterate, primary, upper primary, secondary, senior secondary, or above), socio-economic status of the family (those carrying above or below the poverty line cards or extremely poor people cards or those with no cards), monthly income of the whole family from all sources (in Indian Rupees), study area (rural or urban) and occupation of the head of the family (agriculture, daily wagers, services (private or government or cooperative), business, or unemployed) were asked.

The outcomes in the study included questions about the history of anemia (yes/no/don’t know), consumption of IFA tablets (yes/no), awareness of HIV/AIDS, and sexually transmitted infections/reproductive tract infections (STI/RTI) (yes/no), the practice of open defecation and handwashing after defecation (yes/no), and consumption of 3 or more meals per day (yes/no). We also asked a question pertaining to the practice of safe menstrual hygiene from adolescent girls. Three questions were asked, including the type of sanitary product used during periods (cloth, pad, or both), frequency of the change of pads in a day (three or more times or less than three times a day), and were cloths washed with soap and dried in the sun if reused (yes/no). The safe practices were assessed as use of pads with change frequency of 3 or more times a day or use of cloths with changed frequency three or more than three times a day and washed and dried in sun if reused. Girls with all the safe menstrual hygiene practices were labeled as yes, those with none of the safe practices were labeled no, and the third category was of mixed responses as a mixed category for safe menstrual hygiene practices.

### 2.3. Statistical Analysis

The boy and girl participants were dichotomized as early (10–14 years) and late (15–19 years) adolescents for analysis. The descriptive data were expressed as frequency and percentages (categorical data). Unadjusted and adjusted logistic regression was performed to assess the associations between outcomes (had anemia, consumption of IFA tablets, heard about HIV/AIDS and RTI/STI, open defecation, washed hands after defecation, consumption of 3 or more meals a day, and safe menstrual hygiene practices in girls) and predictors (early age versus late adolescents). In the adjusted regression analysis, we adjusted for other predictors, such as religion, socio-economic status, caste, study area, and education status. Separate analysis was done for boys and girls. The regression analysis data were expressed as unadjusted odds ratio (OR) or adjusted odds ratio (aOR) with 95% confidence interval (95% CI). All the analysis was performed in SPSS version 24.0 (IBM Corp., 145 Armonk, NY, USA). A *p*-value < 0.05 was considered statistically significant.

## 3. Results

As shown in Table 1, the majority of the girls and boys were Hindus. Eighty-six percent of late adolescent girls were Hindu compared to 82.8% of early adolescents. Similarly, 86% of late adolescent boys were Hindu compared to 83% of early adolescent boys. Around 49% of girls and 45% of boys belonged to the scheduled caste/tribe group. Fifteen percent of early adolescent girls and boys belonged to non-marginalized groups compared to 12% of late adolescent girls and 14% late adolescent boys. The most common occupation of the head of the family of boys and girls was daily wagers (50–58%), followed by agriculture (24–32%). Close to two-thirds of girls in the late adolescence group heard of HIV/AIDS; however, only 36% of girls in the early adolescent group heard of HIV/AIDS (Table 2). Similarly, 29.6% of boys in the early adolescence and 55.7% in the late adolescence had heard of HIV/AIDS. Between 23–24% of boys and 28–32% of girls were defecating in the open (Table 2).

Compared to late adolescent girls, early adolescent girls had lower odds of the consumption of IFA tablets (OR (95% CI): 0.78 (0.74–0.83)), hearing about HIV/AIDS (OR (95% CI): 0.34 (0.32–0.36)), or STI/RTI (OR (95% CI): 0.39 (0.36–0.42)), and consumption of 3 or more meals per day (OR (95% CI): 0.86 (0.80–0.93)) (Table 3). However, early adolescent girls had a higher odd of handwashing after defecation (OR (95% CI): 1.29 (1.19–1.41)) safe menstrual practices (OR (95% CI): 1.31 (1.15–1.49)) and lesser odds of open defecation (OR (95% CI): 0.84 (0.79–0.89)) compared to late adolescence girls. These associations remained significant even after adjustments for sociodemographic variables, such as religion, education, socio-economic status, study area, and caste (Table 3).

Similarly, early adolescent boys had lower odds of having heard of HIV/AIDS (OR (95% CI): 0.33 (0.31–0.35)) or RTI/STI (OR (95% CI): 0.47 (0.43–0.50)) and higher odd of handwashing after defecation (OR (95% CI): 1.03 (0.95–1.12)). Contrary to girls, early adolescent boys had higher odds of the consumption of IFA tablets (OR (95% CI): 1.52 (1.43–1.61)) (Table 4). These associations were statistically significant even after adjustments for religion, education, socio-economic status, study area, and caste (Table 4).

Girls in the non-marginalized group had 10 times higher odds of practicing safe menstrual hygiene practices (OR (95% CI): 10.4 (7.6–14.2)), and 3.8 times higher odds of washing hands with soap/ashes after defecation (OR (95% CI): 3.8 (3.1–4.5)) compared to girls belonging to scheduled caste/tribes (Appendix A). Similarly, Hindu girls had a 60% lower probability of practicing safe menstrual hygiene practices (OR (95% CI): 0.4 (0.4–0.5)), and 50% lower probability of washing hands after defecation with soap/ashes (OR (95% CI): 0.5 (0.4–0.6)) compared to non-Hindu girls. However, girls in the non-marginalized groups had a 93% lower chance of defecating in the open (OR (95% CI): 0.07 (0.06–0.09) compared to girls in the scheduled castes/tribe group. And Hindu girls had 2.5 times higher odds of defecating in open (OR (95% CI): 2.5 (2.2–2.8)) compared to non-Hindu girls. These associations were statistically significant in the adjusted model (Appendix A).

Similarly, Hindu adolescent boys had 4.8 times higher odds of defecating in open (OR (95% CI): 4.8 (4.1–5.5)) and 50% lower probability of washing hands after defecation (OR (95% CI): 0.5 (0.4–0.5)) compared to non-Hindu boys (Appendix A). And non-marginalized boys had 2.2 times higher odds of washing hands with soap/ashes after defecation (OR (95% CI): 2.2 (1.9–2.6)) and 88% lower probability of defecating in open (OR (95% CI): 0.12 (0.10–0.14)) compared to their counterparts. These associations were statistically significant in the adjusted model (Appendix A).

## 4. Discussion

Adolescence is a critical stage in the development of adult life, and the behaviors or habits during this phase impact the life course. Between 84–88% of adolescents in our study represented marginalized classes (backward classes or scheduled tribes or castes), 50–60% were below the poverty line or extremely poor, and 94–96% had monthly family income less than INR 10,000 (USD 130). This spotlights the poor socioeconomic background of the study participants.

A higher proportion of adolescent boys (79–80%) and girls (82–84%) in our study had three or more meals in a day compared to a study from Gujarat where only 55% of boys and girls had habits of taking regular (thrice a day) [10]. This study from Gujarat was done on a sample of 1440 school-going adolescents using a self-administered questionnaire. A study among school-going adolescents from Puducherry reported a habit of skipping meals (particularly, breakfast) as common as 23–30%. Furthermore, 26–36% of adolescents skipped meals < 3 times a week and another 8–9% skipped > 3 times a week [22]. Contrary to other studies, Das et al., in their study among 150 adolescents in Himachal Pradesh, reported that most adolescents had three or more than three meals a day (95–96%). Such a difference could be attributed to the small sample size and enrollment of most of the adolescents in the mid-day meal program of the state [23]. The frequency of meal consumption is a crude estimate of dietary intake, and dietary assessment should be done using more precise methods, such as food frequency questionnaire, 24-h dietary recalls, and dietary records. Our adoption of the meal frequency method was primarily due to limited resources and time with the aim to reflect the hunger and not dietary quality [24,25].

In the present study, 23–32% of adolescents practiced open defecation. On the contrary, a multi-centric study from Bihar, Odisha, and Chhattisgarh found that 82% of adolescent girls were practicing open defecation [7]. The national health survey reported that 39% of households practiced open defecation [4]. This difference in the results could be due to the study areas, as we included both urban and rural areas compared to the multi-centric study focusing primarily on the most backward states of India and mainly rural populations. Open defecation poses serious risks for adolescent health, including diarrhea and gastroenteritis [7]. Our findings could be linked to the *Swachh Bharat Mission* (Clean India Mission), which aimed to achieve universal sanitation coverage and ensure sustainable open defecation behaviors in the community with access to solid and liquid waste management facilities [26].

The systematic review and meta-analysis on menstrual hygiene management among girls in India reported that the use of cloths is common among girls; however, their cleaning and drying in the sun is a problem. Girls lack water, privacy, and a drying place, and hence, washing and drying pose a great problem [27]. Multiple reasons attributed to the limited use of sanitary napkins include lack of affordability, difficulty in or shyness about obtaining them, and lack of awareness about them [20]. Around 7–36% of adolescent girls in rural or urban areas do not take baths regularly. To improve menstrual hygiene practices among girls, they need to be educated, informed about various sanitary products, and counseled on myths and misconceptions that surround menstruation [28].

Compliance to IFA consumption ranged between 41–47% among all adolescents except late adolescent boys. Late adolescent girls had the highest compliance compared to the rest, which might be due to the concentrated efforts by the state and the federal government of supplying IFA tablets in schools to make India anemia-free. Another study among school-going adolescents in urban Puduchchery reported 66% compliance to IFA consumption [29]. Similar to this study, we also found an increased reported prevalence of anemia among late adolescent girls and boys compared to early. The changing dietary habits, nutritional requirements, and the start of periods among girls, lack of awareness about anemia and its prevention, and increased access to outside food facilities, and less dependency on parents are some of the risk factors for increased anemia among late adolescents [29]. However, contrary to our findings, a study from Uttar Pradesh reported 68% consumption of IFA tablets among boys and 56% among girls for less than six months. This study was conducted only among early adolescents (10–14 years) studying in government schools [30]. Compliance with IFA consumption is key to the success of anemia free India program and is heavily dependent on the education by teachers in schools and frontline workers in the communities to adolescents.

Around 88% of the college-going adolescents knew about HIV, but only 30.4% knew about other STIs in a study [31]. These findings are much higher than our results, with 30–66% of adolescents that heard about HIV/AIDS and 13–30% about STI/RTI. A larger percentage of girls in our study were aware of HIV/AIDS and STI/RTI than boys. Adolescence is the age of experimentation and sexual debut. This poses an increased risk of HIV/AIDS and other STI/RTI among adolescents. We need to increase awareness about HIV and other STI, and providing curative or counseling services through community-based programs or facilities, such as Adolescents Friendly Health Services (AFHS), is crucial [31].

Late adolescents had better health outcomes compared to early adolescents except for hygiene and sanitation in our study. A study from two of the most backward states of India, Uttar Pradesh and Bihar, reported decreased odds of receiving IFA tablets among boys between 15–19 years compared to boys between 10–14 years. However, such an association was insignificant among girls [9]. Other studies found that early adolescent girls had a higher probability of compliance to IFA consumption than late adolescent girls. Similar to other studies, we found that awareness about HIV/AIDS or STI/RTI increases with age. Late adolescents (boys or girls) are better aware of such issues than early adolescents [20]. Notably, all the hygiene outcomes in our study were found to be better among early than late adolescents. The two plausible explanations for this paradox could be a higher representation of non-marginalized and a lower percentage of Hindu populations in the early adolescent compared to the late adolescent girls and boys. Our results suggest that non-marginalized and non-Hindu populations had higher odds of practicing safe menstrual hygiene practices and washing hands after defecation, and lower odds of defecating in the open compared to their counterparts. In our study, religion and social caste were independent determinants of hygiene practices among adolescents.

Marginalized populations constituted a major proportion of the study participants in our research. Evidence suggests that marginalized populations have poor health indicators and access to health services [32]. Hence, there is a need to achieve the inclusion of disadvantaged and marginalized groups and communities in research projects and prioritize their issues [33].

### Limitations

The present study results should be interpreted in view of the following limitations. Firstly, the study could not capture reproductive health awareness and nutrition outcomes comprehensively due to the limitation of time and financial resources. However, we made an attempt to select 1–2 questions per domain to capture relevant information. Secondly, the study was conducted primarily to understand the situation of marginalized populations; hence, generalization of the results across all castes/social classes may be challenging. But our study added data pertaining to marginalized sections which are inadequately covered in other studies. The way the questions have been interpreted by the respondents or the data collection by female investigators for boys might have affected the results and could be a potential limitation. However, through our training of the investigators, we tried to keep this bias at a minimum. Investigators were locals and knew the local language and dialects, which minimized bias in the way questions were asked and data were obtained within or outside their homes with little disturbance due to noise. Adolescents were free to question investigators and obtain clarifications. In addition, when case investigators encountered problems while obtaining data from boys, male members of the field team supported them. Lastly, there could be issues with the responses of two questions, primarily history of anemia and monthly family income. There could be an underestimation of the responses to these questions because the history of anemia was based on testing and diagnosis by a doctor and recall bias, and adolescents may not know the actual family income.

## 5. Conclusions

The present study concludes that a large proportion of adolescent girls and boys did not consume IFA tablets, had low reproductive health awareness, and defecated in the open. Nearly one-fifth of adolescents did not consume even three meals in a day. The practice of safe menstrual hygiene practices among girls was also limited. Furthermore, early adolescence is a crucial period and is vulnerable to poor health, nutrition, and hygiene outcomes as they had lower odds of consumption of IFA tablets and meals and lower reproductive health awareness than late adolescents. Early adolescents are not represented in most of the national surveys, and, hence, we lack adequate and comprehensive data to support interventions. There is a dire need to start public health interventions from early adolescence for long-term benefits throughout adolescence.

India’s national adolescent health program based on the peer education approach needs to be strengthened for early and late adolescent groups. The adolescent-friendly health services (AFHS) at health centers and the outreach activities to reach adolescents and educate or counsel them on various issues warrant more investments of resources from the government. Besides, schools as a platform to spread health, nutrition, and hygiene awareness have not been tapped to the fullest, which can be achieved under the School Health Awareness Program of the government. Parents, peers, and school teachers are influencers in determining adolescents’ behavior, who ought to be involved through community-based health education and promotion interventions. Evaluation of these programs or interventions needs to be performed simultaneously to assess the effectiveness of a school or community-based adolescent health programs.

## Figures and Tables

**Table 1 healthcare-09-00980-t001:** Distribution of socio-demographic indicators among early and late adolescent boys and girls.

Indicators	Adolescent Girls (*n* = 19,162) *n* (%)		Adolescent Boys (*n* = 19,008) *n* (%)	
	Early Adolescence (10–14 Years) (*n* = 8606) *n* (%)	Late Adolescence (15–19 Years) (*n* = 10,556) *n* (%)	Early Adolescence (10–14 Years) (*n* = 10,412) *n* (%)	Late Adolescence (15–19 Years) (*n* = 8596) *n* (%)
Religion Hindu Non-Hindu *	7132 (82.8) 1474 (17.1)	9105 (86.3) 1451 (13.7)	8646 (83.0) 1766 (17.0)	7425 (86.4) 1171 (13.6)
Caste Non-marginalized Other backward classes Scheduled caste/tribes	1340 (15.6) 3262 (37.9) 4004 (46.5)	1269 (12.0) 3924 (37.2) 5363 (50.8)	1619 (15.5) 4120 (39.6) 4673 (44.9)	1216 (14.1) 3475 (40.4) 3905 (45.4)
Cards (Socio-economic status) Antayodya Ann Yojna *⁑* Below Poverty Line Above Poverty Line No cards	305 (3.5) 4508 (52.4) 3034 (35.3) 759 (8.8)	583 (5.5) 5734 (54.3) 3488 (33.1) 751 (7.1)	330 (3.2) 4882 (46.9) 4287 (41.2) 913 (8.8)	312 (3.6) 4125 (48.0) 3696 (43.0) 463 (5.4)
Education status Illiterate Primary Upper primary Secondary Senior secondary and above	194 (2.3) 2614 (30.4) 4505 (52.3) 1175 (13.7) 118 (1.3)	406 (3.8) 835 (7.9) 2394 (22.7) 3406 (32.3) 3515 (33.3)	114 (1.1) 8808 (84.6) 1305 (12.5) 167 (1.6) 18 (0.1)	175 (2.0) 2515 (29.3) 2946 (34.3) 2050 (23.8) 910 (10.5)
Monthly family income in Indian Rupees ≤5000 INR 5001–10,000 INR >10,000 INR	4180 (48.6) 4109 (47.7) 317 (3.7)	5514 (52.2) 4606 (43.6) 436 (4.1)	4967 (47.7) 4915 (47.2) 521 (5.0)	4074 (47.5) 3993 (46.5) 517 (6.0)
Occupation of the head of the family Agriculture Daily wage earners Services (government, private, or cooperative) Business Unemployed	2012 (23.4) 4939 (57.4) 1021 (11.9) 564 (6.6) 70 (0.8)	2633 (24.9) 6248 (59.2) 954 (9.0) 642 (6.1) 79 (0.7)	3173 (30.5) 5391 (51.8) 1141 (11.0) 658 (6.3) 49 (0.8)	2974 (34.6) 4220 (49.1) 832 (9.7) 507 (5.9) 63 (0.7)
Study area Rural Urban	6035 (70.1) 2571 (29.9)	8555 (81.0) 2001 (19.0)	7852 (75.4) 2560 (24.6)	7188 (83.6) 1408 (16.4)

* Non-Hindu includes Muslim, Buddhist, Jain, Sikhs, Christians, and Parsis. ⁑ Antayodya Ann Yojna includes extremely poor people.

**Table 2 healthcare-09-00980-t002:** Distribution of health, nutrition, and hygiene outcomes among early and late adolescent girls and boys.

Variables	Adolescent Girls (*n* = 19,162)		Adolescent Boys (*n* = 19,008)	
	Early Adolescents (10–14 Years) *n* = 8606; *n* (%)	Last Adolescents (15–19 Years) *n* = 10,556; *n* (%)	Early Adolescents (10–14 Years) *n* = 10,412; *n* (%)	Last Adolescents (15–19 Years) *n* = 8596; *n* (%)
Had anemia ever Yes Don’t know No	776 (9.0) 791 (9.2) 7039 (81.8)	1734 (16.4) 604 (5.7) 8218 (77.9)	784 (7.5) 944 (9.1) 8684 (83.4)	698 (8.1) 612 (7.1) 7286 (84.8)
Consumed IFA tablets Yes No	3566 (41.4) 5040 (58.6)	5000 (47.4) 5556 (52.6)	4333 (41.6) 6079 (58.4)	2741 (31.9) 5855 (68.1)
Heard of HIV/AIDS Yes No	3150 (36.6) 5456 (63.4)	6619 (62.7) 3937 (37.3)	3078 (29.6) 7334 (70.4)	4787 (55.7) 3809 (44.3)
Heard of STI/RTI Yes No	1249 (14.5) 7357 (85.5)	3172 (30.0) 7384 (70.0)	1303 (12.5) 9109 (87.5)	2010 (23.4) 6586 (76.6)
Open defecation Yes No Missing (*n*)	2416 (28.2) 6164 (71.8) 26	3346 (31.8) 7182 (68.2) 28	2516 (24.2) 7896 (75.8) 0	2011 (23.4) 6585 (76.6) 0
Washed hands after defecation with soaps/ashes Yes No	7612 (88.4) 994 (11.6)	9029 (85.5) 1527 (14.5)	9046 (86.9) 1366 (13.1)	7436 (86.5) 1160 (13.5)
Had 3 or more meals in a day Yes No	7029 (81.7) 1577 (18.3)	8840 (83.7) 1716 (16.3)	8253 (79.3) 2159 (20.7)	6773 (78.8) 1823 (21.2)
Safe menstrual hygiene practices Yes Mixed No No periods	977 (11.3) 2915 (33.8) 486 (5.6) 4228 (49.0)	2189 (20.7) 6835 (64.8) 1429 (13.5) 103 (1.0)	-	-

Abbreviations: IFA: Iron Folic Acid tablets, RTI/STI: Reproductive Tract Infections/Sexually Transmitted Infections.

**Table 3 healthcare-09-00980-t003:** Unadjusted and adjusted regression analysis between age groups and health, nutrition, and hygiene outcomes among adolescent girls.

Outcomes	Early Adolescents (10–14 Years) ^¶^		Early Adolescents (10–14 Years) ^¶^	
	Unadjusted Odds Ratio (95% CI)	*p*-Value	Adjusted Odds Ratio * (95% CI)	*p*-Value
Had anemia ever Yes Don’t know No	0.52 (0.47–0.57) 1.53 (1.37–1.70) Reference	<0.001 <0.001	0.48 (0.43–0.54) 1.36 (1.19–1.56) Reference	<0.001 <0.001
Consumed IFA tablets Yes No	0.78 (0.74–0.83) Reference	<0.001	0.84 (0.78–0.90) Reference	<0.001
Heard of HIV/AIDS Yes No	0.34 (0.32–0.36) Reference	<0.001	0.50 (0.47–0.54) Reference	<0.001
Heard of STI/RTI Yes No	0.39 (0.36–0.42) Reference	<0.001	0.52 (0.48–0.57) Reference	<0.001
Open defecation Yes No	0.84 (0.79–0.89) Reference	<0.001	0.90 (0.83–0.98) Reference	0.02
Washed hands after defecation Yes No	1.29 (1.19–1.41) Reference	<0.001	1.52 (1.37–1.68) Reference	<0.001
Had 3 or more meals in a day Yes No	0.86 (0.80–0.93) Reference	<0.001	0.85 (0.77–0.93) Reference	0.001
Safe menstrual hygiene practices Yes Mixed No	1.31 (1.15–1.49) 1.25 (1.12–1.40) Reference	<0.001 <0.001	1.42 (1.23–1.64) 1.37 (1.21–1.55) Reference	<0.001 <0.001

Abbreviations: IFA: Iron Folic Acid tablets, RTI/STI: Reproductive Tract Infections/Sexually Transmitted Infections. * Adjusted for religion, education, socio-economic status, study area, and caste. ^¶^ Reference: Late adolescents (15–19 years).

**Table 4 healthcare-09-00980-t004:** Unadjusted and adjusted regression analysis between age groups and health, nutrition, and hygiene outcomes among adolescent boys.

Outcomes	Early Adolescents (10–15 Years) ^¶^		Early Adolescents (10–15 Years) ^¶^	
	Unadjusted Odds Ratio (95% CI)	*p*-Value	Adjusted Odds Ratio * (95% CI)	*p*-Value
Had anemia ever Yes Don’t know No	0.94 (0.84–1.04) 1.29 (1.16–1.43) Reference	0.27 <0.001	0.87 (0.76–0.99) 1.14 (1.00–1.30) Reference	0.04 0.04
Consumed IFA tablets Yes No	1.52 (1.43–1.61) Reference	<0.001	1.62 (1.49–1.77) Reference	<0.001
Heard of HIV/AIDS Yes No	0.33 (0.31–0.35) Reference	<0.001	0.51 (0.47–0.55) Reference	<0.001
Heard of STI/RTI Yes No	0.47 (0.43–0.50) Reference	<0.001	0.59 (0.54–0.65) Reference	<0.001
Open defecation Yes No	1.04 (0.97–1.11) Reference	0.21	0.92 (0.84–1.01) Reference	0.07
Washed hands after defecation Yes No	1.03 (0.95–1.12) Reference	0.44	1.35 (1.22–1.49) Reference	<0.001
Had 3 or more meals in a day Yes No	1.02 (0.96–1.10) Reference	0.42	1.0 (0.92–1.09) Reference	0.92

Abbreviations: IFA: Iron Folic Acid tablets, RTI/STI: Reproductive Tract Infections/Sexually Transmitted Infections. * Adjusted for religion, education, socio-economic status, study area, and caste. ^¶^ Reference: Late adolescents (15–19 years).

## Data Availability

Data cannot be provided with the paper but can be made accessible on individual request.

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
