# Peer review of "Comparing Reproductive Health Awareness, Nutrition, and Hygiene among Early and Late Adolescents from Marginalized Populations of India: A Community-Based Cross-Sectional Survey"

_healthcare, 2021, doi:10.3390/healthcare9080980_

Round 1
Reviewer 1 Report
This is clearly an important and much needed research study with serious implications for the population of India. With that said, the abstract was not clear (e.g. the use of the term insults makes little sense) and the presentation of the findings without enough context made it jarring and confusing to read.
I would have liked to see additional literature describing adolescent health as it relates to the Indian population. The methods and data were clearly articulated, however the limitations of the research must be expanded on as there are obvious issues related to the way the questions may have been interpreted. The fact that only several questions were asked is also potentially problematic, though nothing can be done at this point but to acknowledge it.
Finally, the bulk of this paper is on the presentation of the data findings; but it then ends rather abruptly without any real conclusion beyond...this is a dire problem in Indian among youths. I would like to see a future research section or a better developed conclusion section with what the next steps should be for schools and or health or government agencies to address these issues.
Author Response
Reviewer 1:
Comment 1#: With that said, the abstract was not clear (e.g. the use of the term insults makes little sense) and the presentation of the findings without enough context made it jarring and confusing to read.
Response: Thanks to the reviewer for highlighting this. I have now edited the abstract, removed the word ‘Insults’ (Line 10, 11 on Page 1) and added odds ratio to make the abstract more precise (Line 19-25 on Page 1).
Comment 2#: I would have liked to see additional literature describing adolescent health as it relates to the Indian population.
Response: Yes, I have added a para describing additional literature about adolescent health, especially among marginalized populations on Page 2 (Lines 65-72).
Comment 3#: The methods and data were clearly articulated, however the limitations of the research must be expanded on as there are obvious issues related to the way the questions may have been interpreted. The fact that only several questions were asked is also potentially problematic, though nothing can be done at this point but to acknowledge it.
Response: I completely agree with the reviewer. I have edited the limitations sections and added the suggested part in the limitations section (Line 304-312; Page 9). “The way the questions have been interpreted by the respondents or the data collection by female investigators for boys might have affected the results and could be a potential limitation. However, through our training of the investigators, we tried to keep this bias at a minimum. Investigators were locals and knew the local language and dialects, which minimized bias in the way questions were asked and data were obtained within or outside their home with little disturbance due to noise. Adolescents were free to question investigators and obtain clarifications. Also, in case investigators encountered problems while obtaining data from boys, male members of the field team supported them.”
Comment 4#: Finally, the bulk of this paper is on the presentation of the data findings; but it then ends rather abruptly without any real conclusion beyond...this is a dire problem in Indian among youths. I would like to see a future research section or a better developed conclusion section with what the next steps should be for schools and or health or government agencies to address these issues.
Response: Yes, the conclusions added abruptly. Thanks for highlighting this. I have now added a paragraph in the conclusion section (Page 10, Line 328-338).
“India’s national adolescent health program based on the peer education approach needs to be strengthened. The adolescent friendly health services (AFHS) at health centers and the outreach activities to reach adolescents and educate or counsel them on various issues warrant more investments of resources from the government. Besides, schools as a platform to spread health, nutrition, and hygiene awareness has not been tapped to the fullest, which can be achieved under the School Health Awareness Program of the government. Parents, peers, and school teachers are influencers in determining adolescents’ behavior, who ought to be involved through community-based health education and promotion interventions. Evaluation of these programs or interventions needs to performed simultaneously to assess the effectiveness of school or community-based adolescent health programs.”
Reviewer 2 Report
Thanks so much for the opportunity to contribute to this valuable work. The paper is generally written well and is of sound scientific methodology. The introduction gives a clear foundation and footing for the reader to have some basic statistics on the population of study. It also highlights the gap that the paper addresses. However, the study rationale may require some strengthening as it is evident that nearly similar studies have earlier been undertaken.
Abstract: The first line of the abstract is a little miss leading to the reader as it seems that the paper only focuses on the early teens but reading through, its ages 10-19 year olds. The abstract could also benefit more from OR and 95% CI to allow the reader know to what extent there is a reduced or increased risk of the dependent variables. Same applied to the results section. It will be great to have some interesting ORs & 95% CIs included in the results narrative.
Lastly, the limitation section - also requires some elaboration on how the authors/researchers attempted to minimize the limitations so as to allow for dependability and reliability of the findings. Recall biases, the use of female as interviewers for the male... might these have had an impact on the responses?
Author Response
Reviewer 2:
Comment 1#: However, the study rationale may require some strengthening as it is evident that nearly similar studies have earlier been undertaken.
Response: I thank the reviewer for this comment. Yes, we have now added a para in the study rationale on Page 2 (Line 63-76).
“Among all social classes, marginalized populations (scheduled castes, or tribes, or other backward classes) are more vulnerable to poor health practices, nutritional in-takes, and hygiene maintenance [14,15]. This is evident from the national surveys reporting 7% teenage pregnancies among tribal populations compared to 1% among non-tribals [16]. Tribals constitute approximately 9% and scheduled castes about 16.6% of the total population of India and are socio-economically disadvantaged and disproportionately affected by limited access to health services [17,18]. Studies found poor health and nutrition indicators, such as lower dietary diversity, higher prevalence of anemia, and limited awareness about HIV/AIDS among adolescents belonging to scheduled castes and tribes compared to non-marginalized groups [19,20].
Most of the studies explored the health, nutrition, and hygiene status of late adolescents, and there has been limited literature for early adolescents across low and middle-income countries. Smaller sample sizes, limited geographic coverage and community-based studies, and missing early Vs. late adolescents’ comparison are some of the lacunae in the previous studies.”
Comment 2#: Abstract: The first line of the abstract is a little miss leading to the reader as it seems that the paper only focuses on the early teens but reading through, its ages 10-19 year olds. The abstract could also benefit more from OR and 95% CI to allow the reader know to what extent there is a reduced or increased risk of the dependent variables.
Response: I agree with the reviewer that the sentence was a bit misleading and results have not been presented properly. I have now changed the first line (Line 10-11, page 1) as well as added OR with 95% CI in the abstract (Line 19-25, Page 1).
Comment 3#: Same applied to the results section. It will be great to have some interesting ORs & 95% CIs included in the results narrative.
Response: Yes, I have now added OR with 95% CI in the results section as well.
Comment 4#: Lastly, the limitation section - also requires some elaboration on how the authors/researchers attempted to minimize the limitations so as to allow for dependability and reliability of the findings. Recall biases, the use of female as interviewers for the male... might these have had an impact on the responses?
Response: Thanks for highlighting this. I have now made the necessary changes on Page 9 (Lines 298-311). I have elaborated how we attempted to minimize the limitations of the study.
“Firstly, the study could not capture reproductive health awareness and nutrition out-comes comprehensively due to the limitation of time and financial resources. However, we made an attempt to select 1-2 questions per domain to capture relevant information. Secondly, the study was conducted primarily to understand the situation of marginalized populations; hence, generalization of the results across all castes/social classes may be challenging. But our study added data pertaining to marginalized sections which are inadequately covered in other studies. The way the questions have been interpreted by the respondents or the data collection by female investigators for boys might have affected the results and could be a potential limitation. However, through our training of the investigators, we tried to keep this bias at minimum. Investigators were locals and knew the local language and dialects, which minimized bias in the way questions were asked and data were obtained within or outside their home with little disturbance due to noise. Adolescents were free to question investigators and obtain clarifications. Also, in case, investigators encountered problems while obtaining data from boys, male members of the field team supported them. Lastly, there could be issues with the responses of two questions, primarily history of anemia and monthly family income. There could be an underestimation of the responses to these questions because the history of anemia was based on testing and diagnosis by a doctor and recall bias, and adolescents may not know the actual family income.”.
Reviewer 3 Report
Thanks for the opportunity.
The focus of the paper is to undertake some comparisons between early and late adolescents from marginalised populations in India, in areas such as reproductive health awareness, nutrition, and hygiene.
It is understandable early adolescents had lower odds of awareness about HIV/AIDS and STI, compared to late adolescents. However, it is hard to believe that early adolescents had lower odds of open defecation and higher odds of hand hygiene after defecation and safe menstrual practices for girls. I would like to see this kind of findings supported by other literatures (excluding the possibility of data issues or errors in the analysis).
In general, the findings in Table 2 are consistent to those in Tables 3 and 4 using unadjusted Logistic regression models. It’s interesting to see the findings for “open defecation” and “washed hands after defecation” are different depending on the method of Logistic regression model, namely unadjusted and adjusted. I wonder if the authors could explain it. I was wondering if the higher proportions of the non-marginalized of the early adolescents (particularly for girls) could help explain it.
Regarding the survey methods, I wonder how a subsequent household was selected if the girl or boy was not found at the first attempt. In addition, I wonder why the second visit was not made if the first was not successful? The survey had a multi-stage survey design, I wonder if the nature of the survey design was taken into account in the analysis using SPSS.
Other minor things listed here:
Line 32: “electrical development”?
Line 41: 42% of girls or women?
Line 77: “One-two” or “One or two”?
Line 94: Remove the dot in “. Measures”
Line 98: the education status: I wonder if it refers to the adolescents themselves or the head of the family. Education status is clearly a time dependent variable as it changes with age.
Line 101: “monthly family income”: does it mean per capita or for the whole family?
Line 215: “…compared to early”: I think “among adolescent girls” should be added.
Line 227: “…13-30% STI/RTI”: “about” should be added before “STI/RTI”.
I tend to think the overall picture of adolescent health is concerning for girls and boys and for early and late adolescents. More work has to be done in this area, including but not limited to research, community programmes and health education and promotion. In the context of COVID-19, things can be more challenging and I wish I could encourage you to continue the good work and sustainable community intervention programmes while controlling the impact of COVID-19 on the communities.
Author Response
Reviewer 3
Comment 1#: It is understandable early adolescents had lower odds of awareness about HIV/AIDS and STI, compared to late adolescents. However, it is hard to believe that early adolescents had lower odds of open defecation and higher odds of hand hygiene after defecation and safe menstrual practices for girls. I would like to see this kind of findings supported by other literatures (excluding the possibility of data issues or errors in the analysis).
Response: Yes, this is the case. I have checked the data and looked out for any possibility of error in the analysis, But there is no such error. The two plausible explanations for this paradox could be a higher representation of non-marginalized and a lower percentage of Hindu populations in the early adolescent compared to the late adolescent girls and boys. Our results suggest that non-marginalized and non-Hindu populations had higher odds of practicing safe menstrual hygiene practices and washing hands after defecation, and lower odds of defecating in the open compared to their counterparts. In our study, religion and social caste were independent determinants of hygiene practices among adolescents
Comment 2#: In general, the findings in Table 2 are consistent to those in Tables 3 and 4 using unadjusted Logistic regression models. It’s interesting to see the findings for “open defecation” and “washed hands after defecation” are different depending on the method of Logistic regression model, namely unadjusted and adjusted. I wonder if the authors could explain it. I was wondering if the higher proportions of the non-marginalized of the early adolescents (particularly for girls) could help explain it.
Response: Thanks for highlighting this. This is what I just explained in the previous comment. We added four supplementary tables to show how socio-demographic variables are associated with the outcomes other than age. The association of the religion and social caste with hygiene outcomes are as follows:
“Girls in the non-marginalized group had 10 times higher odds of practicing safe menstrual hygiene practices (OR (95%CI): 10.4 (7.6-14.2)), and 3.8 times higher odds of washing hands with soap/ashes after defecation (OR (95%CI): 3.8 (3.1-4.5)) compared to girls belonging to scheduled caste/tribes (Supplementary Table 1). Similarly, Hindu girls had a 60% lower probability of practicing safe menstrual hygiene practices (OR (95%CI): 0.4 (0.4-0.5)), and 50% lower probability of washing hands after defecation with soap/ashes (OR (95%CI): 0.5 (0.4-0.6)) compared to non-Hindu girls. However, girls in the non-marginalized groups had a 93% lower chance of defecating in the open (OR (95%CI): 0.07 (0.06-0.09) compared to girls in the scheduled castes/tribe group. And Hindu girls had 2.5 times higher odds of defecating in open (OR (95%CI): 2.5 (2.2-2.8)) compared to non-Hindu girls. These associations were statistically significant in the adjusted model (Supplementary Table 2).
Similarly, Hindu adolescent boys had 4.8 times higher odds of defecating in open (OR (95%CI): 4.8 (4.1-5.5)) and 50% lower probability of washing hands after defecation (OR (95%CI): 0.5 (0.4-0.5)) compared to non-Hindu boys (Supplementary Table 3). And non-marginalized boys had 2.2 times higher odds of washing hands with soap/ashes after defecation (OR (95%CI): 2.2 (1.9-2.6)) and 88% lower probability of defecating in open (OR (95%CI): 0.12 (0.10-0.14)) compared to their counterparts. These associations were statistically significant in the adjusted model (Supplementary Table 4)”.
Comment 3#: Regarding the survey methods, I wonder how a subsequent household was selected if the girl or boy was not found at the first attempt. In addition, I wonder why the second visit was not made if the first was not successful?
Response: Since, it was a cross-sectional study we enrolled participants who were found during the first attempt.
Comment 4#: The survey had a multi-stage survey design, I wonder if the nature of the survey design was taken into account in the analysis using SPSS.
Response: Yes, I agree with the reviewer. We have adjusted for the study area (rural Vs. urban) in the model to account for the study design.
Other minor things listed here:
Line 32: “electrical development”?
Response: I have now deleted the word ‘electrical’.
Line 41: 42% of girls or women?
Response: Its girls only and I have changed the word men to boys on Page 1, line 44.
Line 77: “One-two” or “One or two”?
Response: Yes, I changed it to: One or two on Page 2 Line 90.
Line 94: Remove the dot in “. Measures”
Response: Yes, I have now removed the dot.
Line 98: the education status: I wonder if it refers to the adolescents themselves or the head of the family. Education status is clearly a time dependent variable as it changes with age.
Response: It is the education of adolescents. I do agree that education changes with time, that’s why, adjusted for education in the model.
Line 101: “monthly family income”: does it mean per capita or for the whole family?
Response: It is the monthly income of the whole family. I have specified that in the methodology now on Page 3, Line 114.
Line 215: “…compared to early”: I think “among adolescent girls” should be added.
Response: Yes, I have now added it on Page 8 Line 257. “late adolescent girls and boys compared to early”.
Line 227: “…13-30% STI/RTI”: “about” should be added before “STI/RTI”.
Response: Yes. I have now added it.
Round 2
Reviewer 1 Report
The revisions made addressed my initial concerns. I appreciated the concluding paragraph discussing future approaches and the additional literature review contextualizing the problem.